# Novel *Haemocystidium* sp. Intraerythrocytic Parasite in the Flatback (*Natator depressus*) and Green (*Chelonia mydas*) Turtle in Western Australia

**DOI:** 10.3390/pathogens13121112

**Published:** 2024-12-16

**Authors:** Erina J. Young, Rebecca Vaughan-Higgins, Kristin S. Warren, Scott D. Whiting, Gabriele Rossi, Nahiid S. Stephens, Lian Yeap, Jill M. Austen

**Affiliations:** 1Conservation Medicine Program, School of Veterinary Medicine, Murdoch University, Murdoch, WA 6150, Australia; r.vaughan-higgins@murdoch.edu.au (R.V.-H.); k.warren@murdoch.edu.au (K.S.W.); l.yeap@murdoch.edu.au (L.Y.); 2EnviroVet Consultancy, Sunshine Coast, QLD 4561, Australia; 3Marine Science Program, Department of Biodiversity, Conservation and Attractions, Kensington, WA 6151, Australia; scott.whiting@dbca.wa.gov.au; 4Centre for Animal Production and Health, Food Futures Institute, Murdoch University, Murdoch, WA 6150, Australia; g.rossi@murdoch.edu.au (G.R.); n.stephens@murdoch.edu.au (N.S.S.); 5School of Agriculture Science, Murdoch University, Murdoch, WA 6150, Australia; j.austen@murdoch.edu.au

**Keywords:** flatback turtle, green turtle, *Haemocystidium* sp., haemosporidian parasite, haematology, phylogenetic analysis, *Natator depressus*, *Chelonia mydas*, health assessments

## Abstract

Malaria and other haemosporidian parasites are common in reptiles. During baseline health surveys of sea turtles in Western Australia (WA), haemosporidian parasites were detected in flatback (*Natator depressus*) and green (*Chelonia mydas*) turtle erythrocytes during routine blood film examination. 130 blood samples were screened via polymerase chain reaction (PCR), including 105 *N. depressus,* 20 *C. mydas,* and 5 olive ridley turtles (*Lepidochelys olivacea*). A novel *Haemocystidium* sp. was identified, detected exclusively in foraging turtles and not in nesting turtles. The combined prevalence by microscopic and molecular methods was 16.9% (22/130), primarily affecting immature *C. mydas* (77.3%; 17/22). Mature *N. depressus* were also affected (22.7%; 5/22). DNA sequencing of a partial fragment of the mitochondrial cytochrome *b* (*cytb*) gene together with phylogenetic analysis identified two different *Haemocystidium* sp. genotypes, A and B, with genotype A being most prevalent. The phylogenetic analysis showed close genetic relationships to *Haemocystidium* sp. in freshwater and terrestrial turtles, suggesting a shared evolutionary lineage despite ecological differences. Preliminary analysis indicates that this parasite is incidental, as no association between health and parasite presence or grade was detected. This study provides the first formal detection of haemosporidian parasites in sea turtles, contributing essential baseline data while highlighting their evolutionary significance and host–parasite ecological relationships.

## 1. Introduction

Six of the world’s seven sea turtle species are found in Australia and inhabit the state waters of Western Australia (WA) [1]. All six species are listed as threatened fauna and protected under state and commonwealth legislation. Flatback (*Natator depressus*) and green (*Chelonia mydas*) turtles are listed as ‘vulnerable,’ while olive ridley turtles (*Lepidochelys olivacea*) are listed as ‘endangered’ at both state and federal levels. *Natator depressus* is endemic to Australia and is listed as ‘data deficient,’ while *C. mydas* is listed as ‘endangered’ and *L. olivacea* is listed as ‘vulnerable’ by the International Union for the Conservation of Nature [2].

Sea turtles are frequently reported as sentinels of environmental health but there are limited health data on sea turtles in the Indian Ocean, especially *N. depressus* [3,4]. Disease in sea turtles is regarded as a threat to sea turtle conservation, with pathogens not well understood [1,5]. Furthermore, there is concern for anthropogenic influences and increases in infectious emerging disease with a changing climate [6,7].

Haemosporidia (Apicomplexa) are a diverse group of parasites which infect a wide range of vertebrate hosts, including birds, reptiles and mammals [8,9]. This group includes *Plasmodium* spp., the causative agent of malaria and a parasite of critical One Health significance and ecological importance [10,11]. These parasites are typically transmitted by dipteran vectors and have indirect and complex life cycles, predominantly intraerythrocytic but also involving extraerythrocytic stages [12,13]. There has been significant debate regarding the classification of these parasites, particularly those in the Haemoproteidae family. However, recent molecular and phylogenetic research has reported that the reptilian and avian Haemoproteus clades are different and reptilian Haemoproteus have been reclassified into the *Haemocystidium* genus [8,14,15,16]. Despite a paucity of research into reptilian haemosporidia, *Haemocystidium* spp. have been found globally in a number of locations in freshwater turtle species in India, Turkey, South America and Australia, including the recently described *Haemocystidian chelodinae*-like haemoproteidae in the critically endangered Bellinger River snapping turtle (*Myuchelys georgesi*) [16].

Although blood parasites are common in reptiles, haemosporidians have not been reported in sea turtles in the published literature [17]. As reptiles co-evolved with their parasites, haemosporidians are generally considered incidental, although some are reported to cause disease [18,19]. For instance, blood parasite infections in reptiles may lead to anaemia, haemolysis and splenomegaly, particularly under conditions of stress or immunosuppression [13].

This is the first published report of haemosporidian parasites in sea turtles and describes the morphological and genetic characterisation of a novel *Haemocystidium* parasite at the mitochondrial cytochrome *b* (*cytb*) gene. The study also assesses the risk factors and health effects on the host.

## 2. Methods

### 2.1. Study Sites and Animals

This study was part of a larger research project developing health baselines for WA sea turtles but only turtles molecularly screened for haemosporidians via polymerase chain reaction (PCR) were included (*n* = 130) in this investigation. Study sites were located in northwest WA and included Eighty Mile Beach (19.5931° S, 121.2694° E) and Roebuck Bay (18.0585° S, 122.2831° E) foraging grounds, as well as Eighty Mile Beach and Thevenard Island (21.4563° S, 115.0021° E) rookeries (Figure 1). Between 2016 and 2020, health assessments were performed on nesting *N. depressus* from Eighty Mile Beach (*n* = 28) and Thevenard Island (*n* = 1), foraging *N. depressus* from Eighty Mile Beach (*n* = 6) and Roebuck Bay (*n* = 70), foraging *C. mydas* from Eighty Mile Beach (*n* = 19) and Roebuck Bay (*n* = 1), and foraging *L. olivacea* from Eighty Mile Beach (*n* = 3) and Roebuck Bay (*n* = 2) (Table 1). Foraging turtles were captured either by rodeo [20], dip net or beach jump and were manually restrained for blood sampling. Nesting turtles returning to the ocean were captured by hand and restrained on a purpose-built restraining device for blood collection [21]. All sampling was opportunistic.

### 2.2. Health Assessment

A health assessment was performed on each turtle and included a physical examination, blood sample collection and morphometric data collection. The physical examination included an assessment of body condition, mentation, external abnormalities, epibiota, as well as neurological and musculoskeletal examinations. Morphometrics included curved carapace length (CCL) and curved carapace width (CCW). Foraging turtles also underwent carapace-to-tail length (CT) and weight measurements.

### 2.3. Sample Collection

Blood was collected at the start of the clinical examination to avoid any confounding impacts of further animal handling. The dorsocervical sinus area was prepared with 5% aqueous chlorhexidine gluconate/alcohol (Chlorhex C^®^, Jurox, Rutherford, NSW, Australia). A 20–30 mL blood sample was then collected using either a 3.8 cm 18 G needle or a 2.5 cm 21 G needle (depending on turtle size) and transferred into anti-coagulant tubes (MiniCollect^®^ LH/Lithium–Heparin, Greiner Bio-one, Kremsmünster, Austria or Lithium–Heparin LH/1.3 screw cap SARSTEDT, Nümbrecht, Germany, and Lithium–Heparin BD Vacutainer^®^, BD, Plymouth, UK). A subset of turtles had whole blood transferred into 2.5% glutaraldehyde (1:1) for electron microscopy (*n* = 5) to identify intraerythrocytic parasites. All samples were placed in a portable cooler. The 10 mL lithium–heparin (Li–Hep) vacutainer tubes were centrifuged at 1534× *g* for 10 min (E8V LW Scientific Centrifuge, Lawrenceville, GA, USA) within 12 h of collection, and the plasma pipetted off into 500 µL aliquots and refrigerated or frozen for further biochemical and other analyses.

### 2.4. Clinical Pathology

Haematology was performed at Vetpath Laboratory Services (Ascot, WA, Australia) (now Vetnostics) on the Cell-Dyn 3700 (Abbott Diagnostics, Wiesbaden, Germany) for nesting *N. depressus*, and on the Sysmex XN-1000 (Sysmex, Kobe, Japan) for foraging turtles and included haemoglobin (Hb) and erythrocyte count. The haematocrit was determined manually by placing Li–Hep blood in plain glass capillary tubes (Hurst Scientific, Forrestdale, WA, Australia) and centrifuging at 5345 *g* for three minutes in the Haematokrit 20 (Hettich, Tuttlingen, Germany). Mean cell haemoglobin concentration (MCHC), mean cell haemoglobin (MCH), and mean cell volume (MCV) values were subsequently calculated. The WBC count was estimated by counting cells in 10 representative fields under 10× or 40× objectives, using standard calculation methods, and differential counts were performed under 40×. Heterophil to lymphocyte (H:L) ratio was subsequently calculated as a potential stress indicator. Comments regarding cell morphology, haemoparasites, vacuolation, platelet estimation, and any haemolysis were also included in the clinical pathology reports. The number of abnormal cells or degree of change was classified according to the number of abnormal cells per 100× objective as mild (1+, 1–5 cells), moderate (2+, 6–10 cells), and severe (3+, >10 cells). All blood film examinations were performed by a board-certified clinical pathologist at Vetpath Laboratory Services, a NATA (Sydney, NSW, Australia) accredited laboratory.

Plasma biochemistry was performed at Vetpath Laboratory Services on the Beckman Coulter AU680 (Beckman Coulter, Tokyo, Japan), except for lactate dehydrogenase (LDH) which was tested at Western Diagnostic Pathology (Myaree, WA, Australia) using an Advia Chemistry XPT (Siemens, Tarrytown, NY, USA). The biochemical analytes included creatine kinase (CK), aspartate transaminase (AST), alkaline phosphatase (ALP), alanine aminotransferase (ALT), total bilirubin, blood urea nitrogen (BUN), total bile acids, uric acid, glucose, cholesterol, triglyceride, sodium, potassium, chloride, total protein, albumin, globulin, calcium, phosphorus, magnesium, iron, LDH, and glutamate dehydrogenase (GLDH). Calcium to phosphorus (Ca:P), sodium to potassium (Na:K), and albumin to globulin (A:G) ratios were subsequently calculated. All laboratory testing was performed within 72 h of collection.

Some testing was also performed in the field. The haematocrit was determined using duplicate plain glass capillary tubes (Statspin^®^ Microhaematocrit 40 mm untreated glass tubes, Iris, Chatsworth, CA, USA) filled with Li–Hep whole blood and centrifuged at 6900× *g* for three minutes using a ZipCombo Centrifuge (LW Scientific, Lawrenceville, GA, USA). Total plasma solids (TPS) were determined using a refractometer (Brix 0–32% Refractometer, LW Scientific, Lawrenceville, GA, USA).

### 2.5. Microscopy

#### 2.5.1. Light Microscopy

Blood films were prepared using Li–Hep whole blood and stained with Leishman and Wright–Giemsa to examine for the presence of blood parasites. The slides were air-dried and mounted to a coverslip using DePeX mounting medium (Merck Pty. Ltd., Kilsyth, Victoria, Australia). Photomicrograph images of Haemoproteidae parasites observed in blood films were taken at ×1000 magnification using an image acquisition system (DP-SAL, Olympus, Tokyo, Japan) mounted on an Olympus BX50 light microscope (Olympus, Tokyo, Japan).

Following the discovery of intraerythrocytic haemoparasites on light microscopy, all blood films were re-examined for haemoparasites and their presence and grade were recorded as per the methodology reported in the laboratory testing outlined in Section 2.4. Due to the potential for the association of intraerythrocytic vacuolation with haemoparasites, vacuole presence and grade were also recorded.

#### 2.5.2. Electron Microscopy

An electron microscope (FEI Tecnai G2 Transmission Electron Microscope, FEI Company, Hillsboro, OR, USA) at the Department of Primary Industries and Regional Development (DPIRD), Diagnostics and Laboratory Services (DLS) (South Perth, WA, Australia), was utilised to analyse blood samples preserved in glutaraldehyde for potential intraerythrocytic haemoparasites.

### 2.6. DNA Extraction, PCR and Sequencing

Genomic DNA (gDNA) was extracted from whole blood using the MasterPure^TM^ DNA Purification Kit (Epicentre Biotechnologies, Madison, WI, USA), in accordance with the manufacturer’s guidelines. DNA was eluted in 35 μL of TE buffer and stored at −20 °C until use. A 480 bp region of the *cytb* gene was amplified from DNA samples using a nested PCR assay, using the primary primers HaemNF1 (5′-CATATATTAAGAGAATTATGGAG-3′) and HaemNR2 (5′-AGAGGTGTAGCATATCTATCTAC-3′) [22], and the secondary primers HaemF (5′-ATGGTGCTTCGATATATGCATG-3′) and HaemR2 (5′-GCATTATCTGGATGTGATAATGGT-3′) [23]. The PCR assays were performed in 25 μL volumes and contained approximately 50 ng of gDNA, 0.4 μM of each primer, 0.2 μL Kapa Taq (Kapa Biosystems, Wilmington, MA, USA), 2.5 mM dNTPs, 1.5 mM MgCl_2_ and PCR buffer (Kapa Biosystems, Wilmington, MA, USA). A 1 μL aliquot of the primary PCR product was used as the DNA template for the nested round of PCR. Polymerase chain reaction cycling conditions consisted of initial denaturation at 94 °C for 8 min, followed by 40 cycles of denaturation at 94 °C for 30 s, annealing at 50 °C (primary reaction) or 52 °C (secondary reaction) for 30 s, and extension at 72 °C for 45 s, followed by final extension at 72 °C for 10 min. All controls (no template, extraction–reagent blanks and positive PCR controls) produced appropriate PCR results. The amplified PCR products were purified using an in-house filter tip method as previously described [24] and used for sequencing. All PCR amplicons were submitted to the Australian Genome Research Facility (Perth, WA, Australia) for unidirectional sequencing with a selection of isolates chosen for bidirectional sequencing.

### 2.7. Phylogenetic Analysis

Partial *cytb* sequences (420 bp) from sea turtle genomic DNA samples were aligned with other closely related haemosporidia sequences retrieved from GenBank [25] via BLAST [26] searches. Sequences were aligned with MUSCLE [27] and trimmed to remove terminal gaps in Geneious [28] with *Plasmodium homopolare* (CACO582 MT341242) used as an outgroup. The most appropriate nucleotide substitution model (general time reversible (GTR) and gamma + 1) was tested for in MEGA 6 [29], with the maximum likelihood (ML) tree constructed using 1000 bootstrap replicates. Genetic distances were generated in MEGA 6 based on the Tajima–Nei algorithm. The partial *cytb* sequences generated in the present study have been submitted to GenBank under accession PQ671476 (genotype A) and PQ671477 (genotype B).

### 2.8. Statistical Analysis

As no age class and sex classifications exist for *N. depressus*, individuals were classed as mature if they were longer than the smallest satellite-tracked foraging turtle in this study which had migrated for breeding (≥79.8 cm CCL). Additionally, mature *N. depressus* were classified as male if their carapace-to-tail length was more than the smallest satellite-tracked foraging turtle that had migrated for breeding (≥15.0 cm CT). All *C. mydas* were juveniles (<65.0 cm CCL) and classed as immature [30], while *L. olivacea* < 65.0 cm CCL were classified as immature and those ≥ 65.0 cm CCL were classified as mature [31,32].

Health status was determined through a combination of physical examination findings and blood results via comparison with species-specific blood reference intervals [21,33,34]. Individuals with diminished body condition and/or clinical abnormalities and/or blood results likely to impact health were classed as ‘unhealthy,’ otherwise turtles were classed as ‘healthy.’

The prevalence of *Haemocystidium* sp. was calculated separately for molecular and light microscopy techniques, as well as for the overall (combined) prevalence. Cases were considered positive if either technique was positive for *Haemocystidium* sp. and the combined molecular and light microscopy results were used for statistical testing of presence. The *Haemocystidium* grade analysis was based solely on light microscopy results, where severe (3+) and moderate (2+) were combined (assigned a ‘high’ grade), and mild (1+) and absent (0) were combined (assigned a ‘low/absent’ grade). The association between molecular and light microscopy *Haemocystidium* results was also examined.

Associations between the *Haemocystidium* results and the various risk factors including species, life stage (nesting vs. foraging), maturity, sex, location, season, year and month, as well as health status and body condition were examined. Due to the absence of *Haemocystidium* sp. in nesting turtles, most analyses were performed on the foraging turtle group. Differences between *Haemocystidium* presence results and haematological and biochemical blood results, as well as morphometrics, were also examined. Given reported differences in blood values between species and life stage (nesting vs. foraging), blood analyses were performed on subgroups. Owing to the composition of the turtle species groups, morphometric analyses were also species specific. Other haematological parameters potentially associated with *Haemocystidium* presence, such as polychromasia, anisocytosis, erythrocyte vacuolation, and presence of reactive lymphocytes and phagocytic macrophages were also investigated.

Chi-square analysis was used for categorical data to test for association. Fisher’s exact test was used if any observed value in the contingency table was less than five. Odds ratios were performed if a significant difference was detected, and the odds ratios are presented with associated 95% CIs. For numerical data, the Anderson–Darling test was used to assess normality, followed by the Student’s *t*-test (parametric data) or the Mann–Whitney U test (non-parametric data). These numerical analyses were restricted to groups with a sample size ≥5. For parameters with more than two groups, ANOVA (parametric data) or the Kruskal–Wallis test (non-parametric data) was used, followed by post-hoc Tukey’s HSD or the Dunn test with Bonferroni adjustment, respectively. A *p*-value of <0.05 was considered statistically significant.

## 3. Results

### 3.1. Microscopy Observations

During routine smear examination, intraerythrocytic stages of a novel *Haemocystidium* sp. were detected on blood films from sea turtles sampled in northwest WA. No extraerythrocytic stages of the haemoparasite were observed.

The parasite generally occupied polar positions within the erythrocyte, with some exceptions of immature gamonts identified at the sides of the cell. Small vacuoles inside erythrocytes were often observed, representative of small clear circular stages. Singular infection of novel *Haemocystidium* sp. within erythrocytes was typical, with the occasional observation of two gamonts within one cell. Immature and mature gamonts, including microgamonts (male) and macrogamonts (female), were observed in erythrocytes from turtles infected with novel *Haemocystidium* sp. genotypes A and B. The immature gamonts from both *Haemocystidium* sp. genotypes A and B were polymorphic, being either round or ellipsoid and with large central vacuoles resembling doughnuts. Numerous black to brown pigment granules were observed in immature stages of the parasites’ outer rim, either scattered or clumped together (Figure 2A and Figure 3A).

Mature gamonts of novel *Haemocystidium* sp. genotypes A and B represent typical morphological characteristics of male and female stages, and, like the immature forms, were polymorphic in nature, with different sized gamonts detected, representing small and large forms. The microgamonts are circular to oblong, contain a light pink cytoplasm with zero to two vacuoles and contain numerous black to brown pigment granules scattered or grouped into small clusters (Figure 2B and Figure 3B). The macrogamonts contain nuclear regions, are circular to oblong, have a bluish to purple cytoplasm with zero to nine vacuoles and contain numerous black to brown pigment granules scattered or grouped into small clusters (Figure 2C,D and Figure 3C,D). Two gamonts within a single erythrocyte was found to be more common for novel *Haemocystidium* sp. genotype A (Figure 2A) when compared with erythrocytes infected with novel *Haemocystidium* sp. genotype B, with only a few exceptions (Figure 3E).

Transmission electron microscopy (TEM) was undertaken to further investigate the ultrastructural details of the novel parasite (Figure 4). TEM confirmed the presence of intracellular parasitic stages containing electron-dense material and intact membranes within the erythrocyte.

### 3.2. Morphological and Molecular Prevalence of Novel Haemocystidium sp. from Sea Turtles

The overall prevalence of novel *Haemocystidium* sp. in this study was 16.9% (22/130). Polymerase chain reaction detected more positive cases (14.6%; 19/130) than microscopy (10.8%; 14/130) and there was a significant association between microscopy and PCR (*p* < 0.001) (Table 2).

All positive cases were in the foraging group. The prevalence of novel *Haemocystidium* sp. in the foraging group was 21.8% (22/101), comprising 77.3% (17/22) immature *C. mydas* and 22.7% (5/22) mature *N. depressus* (Table 3).

When separated by species, the prevalence in the *C. mydas* group was 85% (17/20) and the prevalence in the *N. depressus* group was 4.8% (5/105), or 6.5% (5/76) in the foraging *N. depressus* group.

### 3.3. Statistical and Risk Factor Analysis

When considering life stage as a risk factor for novel *Haemocystidium* sp., foraging turtles were more likely to be positive than nesting turtles (*p* = 0.004).

In the foraging group, *C. mydas* (*n* = 20) were more likely to be positive than other species (*n* = 81) (OR = 86.1; CI 21.4–478.3; *p* < 0.001). For foraging location, turtles from Eighty Mile Beach (*n* = 28) were more likely to be positive than turtles from Roebuck Bay (*n* = 73) (OR = 31.1; CI 9.5–126.0; *p* < 0.001). For maturity, immature foraging turtles (*n* = 31) were more likely to be positive than mature foraging turtles (*n* = 70) (OR = 19.0; CI 6.3–67.4; *p* < 0.001). Additionally, there was a significant association between *Haemocystidium* sp. presence (*n* = 22) and month (*p* < 0.001) and year (*p* < 0.001). In contrast, sex and season had no influence on *Haemocystidium* sp. presence in foraging turtles.

With regards to health, no statistically significant associations were found between the novel *Haemocystidium* sp. presence or grade and health status or body condition in the foraging group. Of the 130 turtles examined, 13 were classed as ‘unhealthy,’ including one turtle found to be positive for *Haemocystidium* sp. via molecular testing. The turtle exhibited extensive keratin loss and bone exposure of the carapace, with epithelialisation at the periphery of the lesion, mild anaemia (PCV 18 L/L, Hb 46 g/L), and elevated AST (486 U/L). The remaining unhealthy turtles, all negative for *Haemocystidium* sp., presented with external abnormalities such as lacerations, exudative carapace and plastron lesions, dehydration, heavy barnacle burdens and a range of clinically significant blood alterations, including severe anaemia (PCV 6 L/L), hypoalbuminaemia (6 g/L), leukocytosis (49.9 × 10^9^ cells/L), heterophilia (30.94 × 10^9^ cells/L), eosinophilia (8.88 × 10^9^ cells/L), and elevated GLDH (318 U/L).

In the foraging *N. depressus* blood analysis, no significant differences were observed in any haematological or biochemical parameters between the positive and negative cases, including in the mature foraging *N. depressus* group. There was a negative association between *Haemocystidium* sp. presence and erythrocyte vacuoles in foraging turtles. No haemolysis was observed and small sample sizes precluded certain blood parameter analyses, including species-specific analyses for *C. mydas* and grade-based analyses for *Haemocystidium* sp.

### 3.4. Phylogenetic Relationships of Haemocystidium sp.

The phylogenetic relationship of the novel *Haemocystidium* sp. from Australian sea turtles to other Haemosporidia species at the mitochondrial *cytb* gene was analysed using maximum likelihood analysis (Figure 5). Sequencing of a partial fragment of *cytb* gene and genetic analysis identified two different *Haemocystidium* sp. genotypes, *Haemocystidium* sp. genotype A and *Haemocystidium* sp. genotype B. For ease of phylogenetic analysis, the *Haemocystidium* sp. genotypes, represented by turtle isolate T195 from *C. mydas* (genotype A) and turtle isolate T239 from *N. depressus* (genotype B), were included in the tree. The sea turtle genotypes grouped together within the reptilian clade, together with nominal species, *Haemoproteus caucasica* and *Haemoproteus anatoilicum*, from tortoises (*Testudo graeca* and *Testudo horsfieldii*) and other *Haemocystidium* sp. isolated from African turtles, including the West African mud turtle (*Pelusios castaneus*) and the forest hinge-back tortoise (*Kinixys erosa*).

A distance similarity matrix generated using Tajima–Nei at the *cytb* gene showed that the novel sea turtle *Haemocystidium* sp. genotypes exhibited a 97.0% genetic similarity to each other with twelve single nucleotide polymorphism (SNP) differences across the 420 bp sequence. The sea turtle isolates should not be considered as separate species, as it has been previously reported that, at the *cytb* gene, a genetic distance of 5% or greater is required for species status [14].

The sea turtle *Haemocystidium* sp. genotypes were found to be grouped within the reptilian clade with other *Haemocystidium* species from river turtles. Sea turtle *Haemocystidium* genotype A exhibited the closest genetic similarity, 99.0%, to *Haemocystidium* sp. HO-01 (KX148090), *Haemocystidium* sp. NG-246 (KX148083), *Haemocystidium* sp. OI-87 (KX148084), *Haemocystidium* sp. OI-88 (KX148088) and *Haemosporida* sp. HO-9 (KT367843) from *Kinixys erosa*, as well as to *Plasmodium* sp. 110 (KY631966). Meanwhile, *Haemocystidium* sp. OL-144 (KX148089)*, Haemocystidium* sp. WN-376 (KX148085), *Haemocystidium* sp. OM-429 (KX148087), and *Haemocystidium* sp. OM-412 (KX148086) exhibited a genetic similarity of 98.8% to sea turtle *Haemocystidium* genotype A. Sea turtle *Haemocystidium* genotype B was more distant, with a 96.5% genetic similarity to the same four undescribed *Haemocystidium* sp. HO-01, NG-246, OI -87, OI-88; to *Haemosporida* sp. HO-9 from *Kinixys erosa;* and to *Plasmodium* sp. 110. Additionally, it exhibited a 96.2% similarity to *Haemocystidium* sp. OL-144, WN-376, OM-429 and OM-412.

To investigate the genetic relationship of *Haemocystidium* isolates from other Australian turtles, genetic comparisons were made between *Haemocystidium* sp. from the sea turtles to *Haemocystidium* isolates (*Haemocystidium chelodinae*-like genotype 1 MN316538 and *Haemocystidium chelodinae*-like genotype 2 MN316537) from the Australian Bellinger River snapping turtle (*Myuchelys georgesi*). Sea turtle *Haemocystidium* genotype A exhibited a genetic similarity of 97.3% to both *Haemocystidium chelodinae*-like genotypes 1 and 2 from *Myuchelys georgesi*, while genotype B from the sea turtle exhibited a 95.7% and a 95.1% genetic similarity to *Haemocystidium chelodinae*-like genotypes 2 and 1, respectively.

When comparing sea turtle *Haemocystidium* sp. genotype A with known described Haemosporidia species from tortoises, such as *Haemoproteus anatolicum* isolate 5315 (KM068153) and *Haemoproteus caucasica* isolate 4111 (KM015455), genetic similarities of 97.3% and 96.0% were exhibited, respectively. Sea turtle *Haemocystidium* sp. genotype B, on the other hand, exhibited a 96.0% similarity to *Haemoproteus anatolicum* (isolate 5315 (KM068153) and a 94.0% genetic similarity to *Haemoproteus caucasica* isolate 4111 (KM015455).

## 4. Discussion

During baseline health surveys in WA sea turtles, a novel intraerythocytic *Haemocystidium* sp. parasite was discovered in both *N. depressus* and *C. mydas*. Although Trocini [35] first discovered haemosporidial parasites in WA loggerhead turtles (*Caretta caretta*), this research was not published and these parasites are not recognised as a pathogen of sea turtles [17]. Furthermore, previous research reporting intraerythrocytic protozoal vacuoles has been disputed due to misidentification or lack of evidence [36].

The presence of haemosporidial parasites in only foraging turtles in our study contrasts with Trocini [35], who detected haemosporidial parasites in nesting *C. caretta* (3.1%; 5/161) but not in foraging *C. caretta* (*n* = 72). It is possible that the parasite is site-specific to foraging grounds such as Eighty Mile Beach and Roebuck Bay. Turtles from these areas may have been underrepresented at nesting beaches, potentially explaining the absence of parasites in nesting turtles.

Preliminary analysis indicates no association between *Haemocystidium* sp. presence or grade and health. This finding further supports Trocini’s conclusion that the parasite is incidental, as nesting turtles need to be in good health to complete migration and breeding [36]. The observed lack of an association between presence or grade and health suggests it is well adapted to its host, likely due to co-evolution [37]. However, an underlying health threshold may allow only healthy, parasite-free *N. depressus* to migrate and nest successfully, potentially explaining the absence of parasites in nesting *N. depressus* and contrasting with Trocini’s findings in *C. caretta* [35].

No clinically significant differences were detected in *N. depressus* for blood analytes and presence of *Haemocystidium* sp. parasite. Eosinophils and basophils are frequently reported to be associated with parasitism, but we detected no association between these granulocytes and *Haemocystidium* presence [38,39]. Blood parasites have the potential to cause haemolysis, however no haemolysis was reported in this study, including in *Haemocystidium*-positive cases. Blood parasites can also contribute to anaemia, but the single *Haemocystidium*-positive unhealthy *C. mydas* with anaemia also had chronic carapace lesions and elevated AST, suggesting tissue damage [17]. Hence, the observed anaemia could not be definitively attributed to parasite presence.

As both Trocini [35] and our study found haemosporidial parasites in WA sea turtles, this suggests a regional distribution of the parasite. Although the prevalence of haemosporidial parasites was higher at the Eighty Mile Beach when compared with Roebuck Bay, this may be because Eighty Mile Beach is an undeveloped site, sometimes used as a reference site, while Roebuck Bay is adjacent to the regional town of Broome. Both foraging sites have muddy intertidal zones with mangrove lined tidal inlets, where the animals may be exposed to potential vectors while foraging, basking at the water’s edge, or in-water basking (especially common in *N. depressus*) [40,41]. Possible transmission routes include haemophagous dipteran vectors such as sandflies or mosquitos, which are prevalent in these habitats, or, alternatively, through haematophagous aquatic organisms, including crustaceans, such as amphipods; isopods; or annelids, such as leeches [12,42,43]. Incidentally, no leeches or leech eggs were detected during examination of sea turtles in this study.

Only one *N. depressus* tested positive for *Haemocystidium* sp. via light microscopy, despite multiple positive results through PCR. This may indicate that parasitaemia in *N. depressus* is too low for microscopic detection. Contributing factors may include species-specific differences in immune responses, vector exposure, ecological behaviour, or infection phase. For example, low levels of circulating parasites in *N. depressus* may reflect a chronic phase driven by coevolution and host adaptation, while higher levels in *C. mydas* could indicate an acute phase [13,37,44]. Age-related factors might also play a role, as all *C. mydas* were immature, and may have been more susceptible to higher parasite burdens when compared with the mature *N. depressus* in this study.

Morphologically, *Haemocystidium* sp. in both *C. mydas* and *N. depressus* exhibited characteristics consistent with other reptilian *Haemocystidium* species, including vacuole formation and pigment granule distribution within gametocytes. Morphologically the *Haemocystidium* sp. genotypes were polymorphic in nature, with different sized immature macrogamonts and microgamonts detected. Such variation in life cycle forms makes it difficult to distinguish between the different genotypes based on morphology alone. However, general features between the genotypes were noted. Genotype A more frequently presented two gamonts per erythrocyte, while genotype B typically showed only one, suggesting differences in infection dynamics or life cycle stages. Morphological comparison between *Haemocystidium* sp. from Australian sea turtles and closely related *Haemocystidium* sp. from African turtle isolates (HO-01, NG-246, O1-87,O1-88, HO-9, OL-144,WN-376, OM-412) could not be made due to the lack of morphological studies in the African turtle projects [45,46]. The presence of intracellular vacuoles of varying sizes, macrogamonts, microgamonts, and well-defined parasitic membranes, as observed via electron microscopy, however, align with descriptions of genetically similar *Haemocystidium chelodinae*-like species described from Australian fresh water turtles (*Myuchelys georgesi*) [16]. These features may reflect adaptations to marine hosts, while variations in gamont positioning and vacuole formation could indicate differences in how these genotypes interact with hosts’ immune systems or ecological niches, warranting further investigation.

The phylogenetic analyses of *Haemocystidium* genotypes A and B reveal close genetic relationships to *Haemocystidium* sp. found in freshwater and terrestrial turtles, such as *Kinixys erosa*, *Pelusios castaneus* and *Myuchelys georgesi* [16,46,47]. The placement of both sea turtle *Haemocystidium* sp. genotypes within the reptilian clade further supports the monophyletic grouping of this parasite, being derived from a common ancestor [8,14,15,16]. Despite vast ecological differences between marine and terrestrial habitats, the parasites show a high level of genetic similarity, suggesting a broad host range and ability to adapt to diverse environments. Both genotypes were found in *C. mydas* and *N. depressus*, though genotype B was more prevalent in *N. depressus*, which is endemic to Australia. The restricted geographic range of *N. depressus,* along with ecological and host-specific factors, may have influenced the genetic divergence and prevalence of genotype B, highlighting the interaction between evolutionary and ecological processes. Genotype A demonstrated high genetic similarity to undescribed *Haemocystidium* species in Africa and to *Haemocystidium chelodinae*-like species in Australia, reflecting a broad geographic distribution and host range. In contrast, the divergence of genotype B may reflect local adaptation to its environment or host [14,16].

### 4.1. Limitations and Recommendations

As with other wildlife research, limited sample size is a challenge. Despite a study sample size of *n* = 130, the disparate dataset, comprising multiple species, life stages and maturity, resulted in small sample group sizes, such as *C. mydas* (*n* = 20) and immature *N. depressus* (*n* = 6). While larger samples sizes, including at nesting beaches, are needed to better assess parasite presence and other factors, smaller datasets, often unavoidable in endangered species studies, can provide valuable exploratory insights [48]. Additional satellite tracking could help reveal breeding migrations, potentially linking nesting sites to foraging areas where infection may occur. Larger studies, including naïve hosts, are essential to clarify the effects of *Haemocystidium* sp. on hosts, the role of infection phase in parasite dynamics, and the lack of correlation with health parameters.

Although molecular techniques are more sensitive when detecting *Haemocystidium* than morphological methods, there were several cases which were found to be positive with microscopy but negative with PCR. It is possible that DNA amplification failed due to low grade parasitemia, reducing the concentration of the starting DNA template or, conversely, high concentrations of DNA may have inhibited PCR amplification due to the presence of nucleated red blood cells in the sample.

Although the novel *Haemocystidium* sp. appears to be incidental, further research is recommended to better understand its vector and transmission routes, including molecular screening of epibiota and ectoparasites collected from the turtles. Additionally, further studies in morphology and phylogenetics are necessary to fully characterise and name the novel *Haemocystidium* species infecting both Australian and African turtles.

Initially, erythrocyte vacuoles were hypothesised to be a proxy for haemoparasites; however, statistical analysis revealed a negative association. While EDTA is typically used to preserve cell morphology, it was avoided in this study, and Li–Hep was used due to its lower likelihood of causing haemolysis and cellular distortion in reptilian RBCs [49,50]. However, Li–Hep can still cause artifacts, such as vacuole-like structures, which may result from the cellular stress associated with storage time, temperature and the effects of the anticoagulant [50,51,52]. To minimise these artefacts, preparing fresh blood smears is recommended for accurate morphological assessment.

Finally, as there are no current age-class classifications in *N. depressus*, there may be some overlap between suspect mature and immature classes. It is important that age-class and sex classifications are developed for *N. depressus* to facilitate more accurate assessments in future studies.

### 4.2. Conclusions

Considering the limited research on sea turtle health in the Indian Ocean, especially for *N. depressus*, the discovery of a novel haemoparasite in this ocean basin is not unexpected. Our findings suggest that this haemoparasitic infection affects multiple species in the region and is considered incidental. As marine diseases are predicted to increase in the future, with changes in environmental conditions, such as increased temperatures, predicted to favour conditions for parasites, it is critical to develop disease baselines now. Identifying infections of potential importance in threatened species will allow us to establish baseline health data, monitor levels of disease and develop novel ways to mitigate health impacts [53,54,55].

## Figures and Tables

**Figure 1 pathogens-13-01112-f001:**
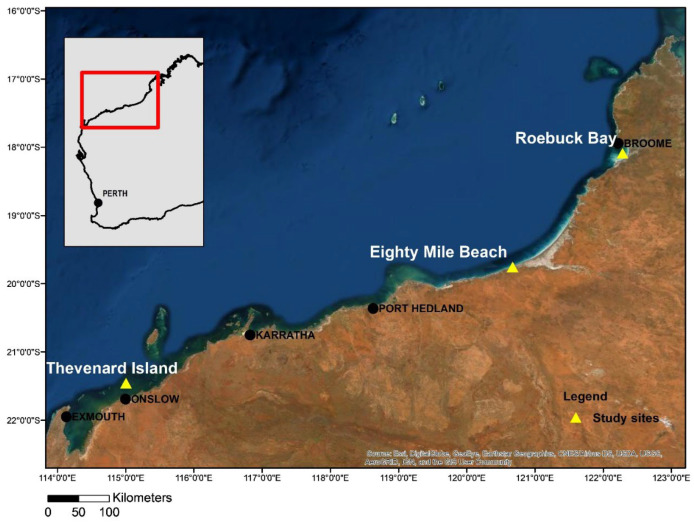
Location of the study sites in northwest Western Australia (WA).

**Figure 2 pathogens-13-01112-f002:**
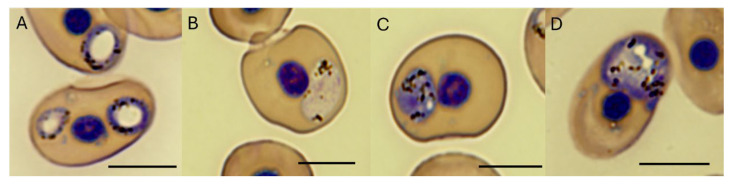
Light photomicrographs of novel *Haemocystidium* sp. genotype A, isolated from Australian sea turtles. (**A**) Immature gamonts of *Haemocystidium* sp., with a large vacuole and granules in the outer rim of the parasite from *Chelonia mydas*. (**B**) Oblong light pink microgamont of *Haemocystidium* sp., with granules scattered and clumped together within the parasite from *C. mydas.* (**C**) Small round dark blue macrogamont of *Haemocystidium* sp., with a purple nuclear region, single vacuole and scattered granules within the parasite from *C. mydas.* (**D**) Large round dark blue macrogamont of *Haemocystidium* sp., with vacuoles and scattered granules within the parasite from *C. mydas.* Scale bar represents 10 μm.

**Figure 3 pathogens-13-01112-f003:**
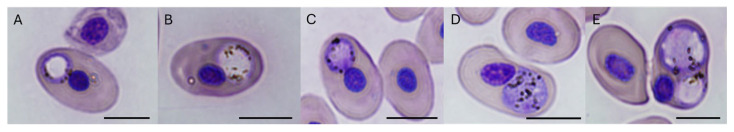
Light photomicrographs of *Haemocystidium* sp. genotype B, isolated from Australian sea turtles. (**A**) Immature gamonts of *Haemocystidium* sp., with a large vacuole and granules in the outer rim of the parasite from *C. mydas.* (**B**) Light pink microgamont of *Haemocystidium* sp., with granules clumped and scattered together within the gamont from *C. mydas.* (**C**) Small purple macrogamont of *Haemocystidium* sp., containing a nuclear region and granules within the parasite from *C. mydas.* (**D**) Large purple macrogamont of *Haemocystidium* sp., containing a nuclear region, granules and vacuoles within the parasite from *C. mydas*. (**E**) Two gamonts of *Haemocystidium* sp., causing nuclear displacement within an erythrocyte from *C. mydas.* Scale bar represents 10 μm.

**Figure 4 pathogens-13-01112-f004:**
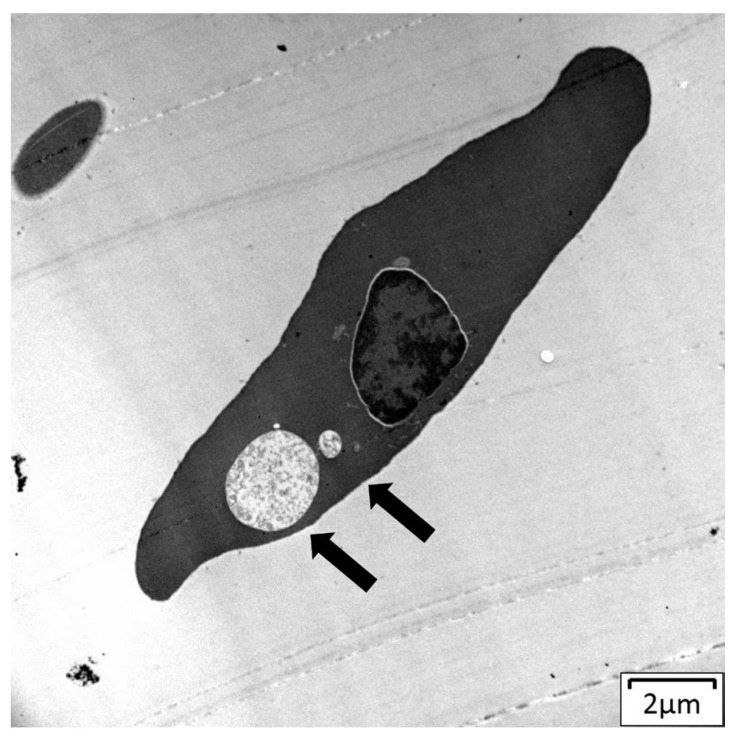
Transmission electron microscopy (TEM) of a novel *Haemocystidium* sp. parasite within an erythrocyte of a *Natator depressus,* showing electron-dense intracellular parasitic stages (arrows).

**Figure 5 pathogens-13-01112-f005:**
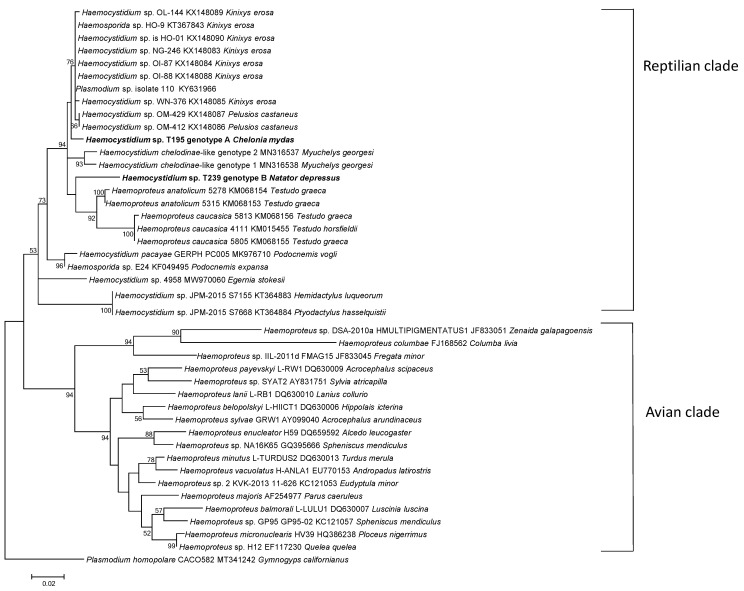
Phylogenetic relationships of *Haemocystidium* sp. genotypes A and B isolated from Australian sea turtles to other Haemoproteidae sp. using maximum likelihood analysis, based on partial (420 bp) *cytb* sequences. Percentage support (>50%) from 1000 replicates is indicated at the left of the supported node.

**Table 1 pathogens-13-01112-t001:** Demographics and distribution of sea turtle species across study sites.

Life Stage	Species	Maturity	Sex	Eighty Mile Beach	Roebuck Bay	Thevenard Island
*Foraging*	*N. depressus*	Immature	Female	-	2	-
*Foraging*	*N. depressus*	Immature	Indeterminate	-	4	-
*Foraging*	*N. depressus*	Mature	Female	3	39	-
*Foraging*	*N. depressus*	Mature	Male	3	25	-
*Foraging*	*C. mydas*	Immature	Indeterminate	19	1	-
*Foraging*	*L. olivacea*	Immature	Indeterminate	2	1	-
*Foraging*	*L. olivacea*	Mature	Female	1	1	-
*Nesting*	*N. depressus*	Mature	Female	28	-	1

**Table 2 pathogens-13-01112-t002:** Cross-tabulation of microscopy and PCR *Haemocystidium* results.

	Microscopy +	Microscopy −
PCR +	11	8
PCR −	3	108

**Table 3 pathogens-13-01112-t003:** Sea turtle cases found to be positive for novel *Haemocystidium* sp. by microscopy and/or PCR, including demographics. All were foraging turtles (* WTR211 was the only turtle classed as ‘unhealthy;’ the rest were classed as healthy).

ID	Species	Maturity	Sex	Location	Year	Month	CCL (cm)	Weight (kg)	*Haemocystidium* grade	Microscopy	PCR	Genotype
WTR154	*N. depressus*	Mature	Female	Roebuck Bay	2018	June	87.8	75		Negative	Positive	A
WTR167	*N. depressus*	Mature	Female	Roebuck Bay	2018	June	85.8	73.5		Negative	Positive	B
WTR194	*C. mydas*	Immature	Indeterminate	Eighty Mile Beach	2019	April	49.4	14.75	1+	Positive	Negative	NA
WTR195	*C. mydas*	Immature	Indeterminate	Eighty Mile Beach	2019	April	41	8.25		Negative	Positive	A
WTR196	*C. mydas*	Immature	Indeterminate	Eighty Mile Beach	2019	April	62	27.75	1+	Positive	Positive	A
WTR198	*C. mydas*	Immature	Indeterminate	Eighty Mile Beach	2019	April	49.9	14.1	1+	Positive	Positive	A
WTR199	*C. mydas*	Immature	Indeterminate	Eighty Mile Beach	2019	April	60.5	27.75	1+	Positive	Positive	A
WTR200	*C. mydas*	Immature	Indeterminate	Eighty Mile Beach	2019	April	54	20.5	1+	Positive	Negative	NA
WTR201	*C. mydas*	Immature	Indeterminate	Eighty Mile Beach	2019	April	39.5	6.5		Negative	Positive	A
WTR202	*C. mydas*	Immature	Indeterminate	Eighty Mile Beach	2019	April	41	8	3+	Positive	Positive	A
WTR203	*C. mydas*	Immature	Indeterminate	Eighty Mile Beach	2019	April	55.6	20.25	2+	Positive	Positive	A
WTR204	*C. mydas*	Immature	Indeterminate	Eighty Mile Beach	2019	April	45.6	11.75	1+	Positive	Positive	B
WTR205	*C. mydas*	Immature	Indeterminate	Eighty Mile Beach	2019	April	43.2	10.75	1+	Positive	Positive	B
WTR207	*C. mydas*	Immature	Indeterminate	Eighty Mile Beach	2019	April	47.2	11.75	1+	Positive	Positive	A
WTR208	*C. mydas*	Immature	Indeterminate	Eighty Mile Beach	2019	April	48.4	14.25	1+	Positive	Positive	A
WTR209	*C. mydas*	Immature	Indeterminate	Eighty Mile Beach	2019	April	45.7	13.25		Negative	Positive	A
WTR210	*C. mydas*	Immature	Indeterminate	Eighty Mile Beach	2019	April	47	11	1+	Positive	Positive	A
WTR211 *	*C. mydas*	Immature	Indeterminate	Eighty Mile Beach	2019	April	48.5	12.25		Negative	Positive	A
WTR212	*C. mydas*	Immature	Indeterminate	Eighty Mile Beach	2019	April	42.5	7.75	1+	Positive	Positive	B
WTR215	*N. depressus*	Mature	Female	Eighty Mile Beach	2019	May	88	73	1+	Positive	Negative	NA
WTR226	*N. depressus*	Mature	Male	Roebuck Bay	2019	June	89.3	86.5		Negative	Positive	A
WTR239	*N. depressus*	Mature	Female	Roebuck Bay	2019	June	90.3	90		Negative	Positive	B

## Data Availability

The original contributions presented in the study are included in the article, further inquiries can be directed to the corresponding author.

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
