# Peer review of "Novel *Haemocystidium* sp. Intraerythrocytic Parasite in the Flatback (*Natator depressus*) and Green (*Chelonia mydas*) Turtle in Western Australia"

_pathogens, 2024, doi:10.3390/pathogens13121112_

Round 1
Reviewer 1 Report
Comments and Suggestions for Authors
This study reports prevalence of haemoparasites in the flatback turtles and green sea turtles in Western Australia. Two novel Haemocystidium spp. (Apicomplexa: Haemoproteidae) were molecularly differentiated, and their prevalence, pathogenicity (effects to host health parameters), and infection risk factors were studies. To disclose pathogens of animals of nature conservation, this study is very important. To improve parts of the manuscripts, several suggestions are mentioned below.
1. L2-L4 [title]: Initially, the authors give us an impression that two genotypes (cytb haplotypes) of a single Haemocystidium species were for the first time found in the sea turtles in WA. After reading, these two haplotypes are distinct, suggesting two independent species in them. The initial impression of a single parasite species comes from “Haemocystidium sp.” in the title and abstract. It is better to reorganize the manuscript to clarify the state of isolated pathogens (two Haemocystidium spp.).
2. L19-L20 [abstract]: Modify the sentence. It takes us to a question that informally researchers know it.
3. L22-L23 [abstract]: Without confirmation by transmission electron microscopy (TEM), blood smears gave the authors that intra-erythrocytic organisms were parasites. TEM observation in this study, represented by Figure 4, gave little information to this study. Remove this sentence starting by “Electron”.
4. L25 [abstract]: What is “PCR”? Numbers of examined samples, “130”, include 5 blood samples from the olive ridley turtles (Lepidochelys olivacea). Results are written using this number “130”, and we have given two turtle names. After reading the text, we understand that sample numbers “130” include other turtle blood samples. It is better to rewrite all parts of this abstract to clearly give us the outline of this study.
5. L26-L27 [abstract]: Modify “The combined prevalence (microscopy and molecular)”. For example, “The combined prevalence by microscopic and molecular approaches”.
6. L28 [abstract]: The term “cytb” (in italic) is “cytochrome b gene”, not “cytochrome b”.
The term “cytochrome b” is “CYTB”. Anyway, “b” between “cytochrome” and “gene” is in italic.
7. L30-L33 [abstract]: It is nice to discuss the haemoparasites detected in this study by two species, not two genotypes of a single species. In this sense, it is better to reorganize the abstract and text of this study.
8. L34-L36 [abstract]: Parasites without evident pathogenicity in the hosts are also important as discussed in the text from evolutionary aspects of the parasites. Closing sentence does not reflect the value of this study (the reviewer believes it is big in nature science).
9. L51: “IUCN 2019” not in the Reference list.
10. L68-L74: Unclear sentence. “sp.” means a single species, and “spp. means two or more species.
11. L81: “cytochrome b gene (cytb) locus” where “b” and “cytb” in italic.
12. L87: What is “PCR”?
13. L105-L106 [Table 1]: How to combine “species” and “maturity” is unclear.
14. L110: How could the authors assess “mentation” of turtles? It is just the reviewers’ question.
15. L119-L210 (many sites): Superscribe “®” and “TM”: ®, TM. Basically, “product name (Company name; city, state, country)”, but in the manuscript, the authors mention them variably, namely, full description to simple description like “product (country)”. Keep consistency.
16. L160-L162: What are “BUN” and “LDH”.
17. L220: Bidirectional sequencing is recommended to get long sequences suitable for phylogenetic analysis.
18. L273: “Student’s t-test”, where “t” prior to “-test” in italic.
19. L276: “… adjustment, respectively.” “A p value of less than 0.05 is considered statistically significant.” “p” before “value” in italic. Inclusion of 0.05 or not is not clear by the current expression.
20. L295-L307: Sizes of gamonts of different genotypes are different? Microphotographs of gamonts give such impression. Could you provide four photographs for Figures w and 3? Then double the size of figures by arrangement of two by two to show clearly the parasites. This comment is partially related to the comment #34.
21. L304: “single”, not “singular”
22. L312: “microgamont” (?)
23. L331-L335 [Figure 4]: It is better to focus on the parasite (Lower half of the erythrocyte shown in the current photograph). Contents of a gamont and a vacuole in the erythrocyte seem to be similar. How the authors differentiate them to be two organisms, not fragments of a single orgamism?
24. L336-L348: How about infection intensity? No information. It must be closely related to health condition, or health condition assessment. Differences between haemochemical parameters of heavily infected and lightly to none infected individuals could be analysed.
25. L353-L361: “p” (represent as p-value) in italic.
26. L364-L366: In the section 2.4, numerous parameters for health assessment are shown. But, in the result section, nothing
27. L379, L380: Refer to comments #6 and #11.
28. L380-LL383: Mention about intra-group (genotypes in the present version of manuscript) nucleotide sequence variation. Absolutely identical? Recommend again bi-directional sequencing of all or selected (at least three isolates per genotype).
29. L387: “nominal species” (?)
30. L425-L427: Closing parentheses are lacked? Or, remove unnecessary starting parenthesis.
31. L449-L451: In this manuscript, infection intensity is not shown. Similarly, any changes of haemochemical parameters are mentioned. It is unclear how the authors concluded their results as no association between Haemocystidium infection and health. Intra-erythrocytic stage and extra-erythrocytic stage of the haemoparasite may affect host health. How do the authors understand Hamocystidium infection and its impact to the host? It is unclear in this manuscript.
32. L463-L465: No information about haemolysis evaluation is given in the results section, but here the authors give it.
33. L481-L488: Acute infection phase? Chronic infection phase?
34. L489-L508: Discuss morphological differences of species A and B, which could support molecular differences between them, after writing information in the Results section.
35. L509-L528: It is unclear whether the authors want to discuss about speciation of the species (evolution) or ecological relationships between the species and their distribution.
36. L542-L544: The parasite studies here is a protozoon. It is necessary why the immature stage has limited DNA compared to the developed stage by a reference.
37. L18-L578: To clean-up the study results and discussion of the present study, it is better to re-organize the contents, discard unnecessary descriptions and discussion, and add description of important data. This is significant study, and its improvement increases the scientific value of this study.
38. L680: Remove “-104576”.
39. L690: “343-344”
Author Response
Reviewer 1
This study reports prevalence of haemoparasites in the flatback turtles and green sea turtles in Western Australia. Two novel Haemocystidium spp. (Apicomplexa: Haemoproteidae) were molecularly differentiated, and their prevalence, pathogenicity (effects to host health parameters), and infection risk factors were studies. To disclose pathogens of animals of nature conservation, this study is very important. To improve parts of the manuscripts, several suggestions are mentioned below.
Comments 1: L2-L4 [title]: Initially, the authors give us an impression that two genotypes (cytb haplotypes) of a single Haemocystidium species were for the first time found in the sea turtles in WA. After reading, these two haplotypes are distinct, suggesting two independent species in them. The initial impression of a single parasite species comes from “Haemocystidium sp.” in the title and abstract. It is better to reorganize the manuscript to clarify the state of isolated pathogens (two Haemocystidium spp.).
Response 1: We appreciate the Reviewer's suggestion that the two Haemocystidium genotypes from sea turtles may represent separate species. However, based on the genetic distance between the two genotypes (97%) and also known species being more than 5% at the cytb locus, we conservatively classify them as genotypes rather than separate species. This approach is consistent with the findings of Pineda-Catalan et al. (2013), which established a threshold of 5% or greater genetic distance at the cytb locus for species status.
Furthermore, our previous work on the Bellinger River snapping turtle (Austen et al., 2020) also supports this approach. In that study, we reported that the two H. chelodinae-like genotypes exhibited 98.5% genetic similarity to each other, and we concluded that they should not be considered separate species due to the genetic distance being less than the 5% threshold.
We believe that applying this consistent and evidence-based approach to species classification is essential to avoid over-splitting and ensure that our taxonomic designations are reliable and comparable across studies. Therefore, we maintain our original classification of the two Haemocystidium genotypes as genotypes rather than separate species.
- Austen JM, Hall J, Zahedi A, Goften A, Ryan U. Further characterisation of Haemocystidium chelodinae-like Haemoproteidae isolated from the Bellinger River snapping turtle (Myuchelys georgesi). Parasitol Res. 2020 Feb;119(2):601-609. doi: 10.1007/s00436-019-06547-9. Epub 2019 Nov 21. PMID: 31754857.
- Pineda-Catalan O, Perkins SL, Peirce MA, Engstrand R, Garcia-Davila C, Pinedo-Vasquez M, Aguirre AA. Revision of hemoproteid genera and description and redescription of two species of chelonian hemoproteid parasites. J Parasitol. 2013 Dec;99(6):1089-98. doi: 10.1645/13-296.1. Epub 2013 Sep 13. PMID: 24032642."
The below statement has now been added into the text for clarity.
L467-L471 “The sea turtle isolates should not be considered as separate species, as it has been previously reported that at the cytb gene, a genetic distance of 5% or greater is required for species status (Pineda-Catalan et al. 2013).
The sea turtle Haemocystidium sp. genotypes grouped….”
Comments 2: L19-L20 [abstract]: Modify the sentence. It takes us to a question that informally researchers know it.
Response 2: The sentence in L19–L20 has been removed. To address this concern, changes were made to the final sentence of the abstract (L34–38), which now reads:
L38-L39 "This study provides the first formal detection of haemosporidian parasites in sea turtles”.
Comments 3: L22-L23 [abstract]: Without confirmation by transmission electron microscopy (TEM), blood smears gave the authors that intra-erythrocytic organisms were parasites. TEM observation in this study, represented by Figure 4, gave little information to this study. Remove this sentence starting by “Electron”.
Response 3: Sentence removed as suggested.
Comments 4: L25 [abstract]: What is “PCR”? Numbers of examined samples, “130”, include 5 blood samples from the olive ridley turtles (Lepidochelys olivacea). Results are written using this number “130”, and we have given two turtle names. After reading the text, we understand that sample numbers “130” include other turtle blood samples. It is better to rewrite all parts of this abstract to clearly give us the outline of this study.
Response 4: The term Polymerase Chain Reaction (PCR) has been defined at its first mention, and additional sample size details have been provided to resolve ambiguity. The revised sentence now reads:
L23-L26 "130 blood samples were screened via Polymerase Chain Reaction (PCR) including 105 N. depressus, 20 C. mydas, and 5 olive ridley turtles (Lepidochelys olivacea). A novel Haemocystidium sp. was identified".(LX-Y)
Comments 5: L26-L27 [abstract]: Modify “The combined prevalence (microscopy and molecular)”. For example, “The combined prevalence by microscopic and molecular approaches”.
Response 5: The phrase "The combined prevalence (microscopy and molecular)" has been revised for clarity and now reads:
L27-L28 "The combined prevalence by microscopic and molecular methods was 16.9% (22/130)."
Comments 6: L28 [abstract]: The term “cytb” (in italic) is “cytochrome b gene”, not “cytochrome b”.
The term “cytochrome b” is “CYTB”. Anyway, “b” between “cytochrome” and “gene” is in italic.
Response 6: The term cytochrome b (cytb) gene has been updated for consistency and accuracy. This sentence has also been modified for improved readability and now reads:
L30-L32 “DNA sequencing of a partial fragment of the mitochondrial cytochrome b (cytb) gene together with phylogenetic analysis identified two different Haemocystidium sp. genotypes”.
Comments 7: L30-L33 [abstract]: It is nice to discuss the haemoparasites detected in this study by two species, not two genotypes of a single species. In this sense, it is better to reorganize the abstract and text of this study.
Response 7: As explained in Comment 1 the two Haemocystidium sp. genotypes A and B isolated from Australian sea turtles are not considered as a separate species as they exhibited a 97% genetic similarity to each other which is less than the 5% difference required for species status (Pineda-Catalan et al. 2013). The use of genotypes A and B will therefore be used throughout the manuscript.
Comments 8: L34-L36 [abstract]: Parasites without evident pathogenicity in the hosts are also important as discussed in the text from evolutionary aspects of the parasites. Closing sentence does not reflect the value of this study (the reviewer believes it is big in nature science).
Response 8: We appreciate the reviewer’s emphasis on the importance of this study from an evolutionary and ecological perspective. The closing sentence of the abstract has been revised to better reflect the value of the study, and now reads:
L38-L41 "This study provides the first formal detection of haemosporidian parasites in sea turtles, contributing essential baseline data while highlighting their evolutionary significance and host-parasite ecological relationships."
Comments 9: L51: “IUCN 2019” not in the Reference list.
Response 9: We apologise for the oversight. The IUCN citation has now been added to the References, as follows:
L772-L773 "IUCN. The IUCN Red List of Threatened Species, Version 2024-2. Available online: http://www.iucnredlist.org (accessed on 15 May 2020)."
Comments 10: L68-L74: Unclear sentence. “sp.” means a single species, and “spp. means two or more species.
Response 10: Thank you for pointing this out. The text has been updated for accuracy and now reads:
L79-L81 "Despite a paucity of research into reptilian haemosporidia, Haemocystidium spp. have been documented globally across multiple locations."
Comments 11: L81: “cytochrome b gene (cytb) locus” where “b” and “cytb” in italic.
Response 11: The term cytochrome b gene (cytb) has been updated here and throughout the manuscript. An additional change has been made to this sentence to improve consistency and now reads:
L95-L96 “mitochondrial cytochrome b (cytb) gene, as well as assessing risk factors and health effects on the host.”
Comments 12: L87: What is “PCR”?
Response 12: The term Polymerase Chain Reaction (PCR) has been defined again at first mention in the body of the manuscript:
L101 "Polymerase Chain Reaction (PCR)"
Comments 13: L105-L106 [Table 1]: How to combine “species” and “maturity” is unclear.
Response 13: Thank you for pointing this out. I have edited the table for improved clarity (L119).
Comments 14: L110: How could the authors assess “mentation” of turtles? It is just the reviewers’ question.
Response 14: Thank you for your question. Mentation in turtles was assessed through behavioural observation and responsiveness to stimuli. These assessments followed standard reptile health evaluation protocols which provide insight into the neurological and general health status of an animal.
Comments 15: L119-L210 (many sites): Superscribe “®” and “TM”: ®, TM. Basically, “product name (Company name; city, state, country)”, but in the manuscript, the authors mention them variably, namely, full description to simple description like “product (country)”. Keep consistency.
Response 15: Thank you for your detailed feedback. The loss of superscripts for “®” and “TM” likely occurred during the editing process. These have now been corrected throughout the manuscript. Regarding the consistency in company names, city, state and country, I have reviewed and updated product and laboratory details as follows: Company Name, City, State (if applicable), Country. Below is a list of the changes (L129-L241):
- Chlorhex C® (Jurox, Australia): Updated to Chlorhex C® (Jurox, Rutherford, NSW, Australia).
- MiniCollect® LH/Lithium Heparin (Greiner Bio-one, Austria): Updated to MiniCollect® LH/Lithium Heparin (Greiner Bio-one, Kremsmünster, Austria).
- LH/1.3 screw cap SARSTEDT, Germany: Updated to LH/1.3 screw cap SARSTEDT, Nümbrecht, Germany).
- BD Vacutainer®, BD-Plymouth, UK: Updated to BD Vacutainer®, BD, Plymouth, UK
- E8V LW Scientific Centrifuge (LW Scientific, Lawerence, GA, USA): Updated to E8V LW Scientific Centrifuge (LW Scientific, Lawrenceville, GA, USA).
- Vetpath Laboratory Services (Ascot, WA): Updated to Vetpath Laboratory Services (Ascot, WA, Australia)
- Cell-Dyn 3700 (Abbott Diagnostics, Germany): Updated to Cell-Dyn 3700 (Abbott Diagnostics, Wiesbaden, Germany).
- Sysmex XN-1000 (Sysmex, Japan): Updated to Sysmex XN-1000 (Sysmex, Kobe, Japan).
- Hurst Scientific, Australia: Updated to Hurst Scientific, Forrestdale, WA, Australia
- Haematokrit 20 (Hettich, Germany): Updated to Haematokrit 20 (Hettich, Tuttlingen, Germany).
- NATA (Sydney, NSW): Updated to NATA (Sydney, NSW, Australia)
- Beckman Coulter AU680 (Beckman Coulter, Japan): Updated to Beckman Coulter AU680 (Beckman Coulter, Tokyo, Japan).
- Western Diagnostic Pathology (Myaree, WA): Updated to Western Diagnostic Pathology (Myaree, WA, Australia)
- Advia Chemistry XPT (Siemens, USA): Updated to Advia Chemistry XPT (Siemens, Tarrytown, NY, USA).
- ZipCombo Centrifuge (LW Scientific, Lawerence, GA, USA): Updated to ZipCombo Centrifuge (LW Scientific, Lawrenceville, GA, USA).
- Brix 0–32% Refractometer (LW Scientific, USA): Updated to Brix 0–32% Refractometer (LW Scientific, Lawrenceville, GA, USA).
- FEI Tecnai G2 Electron Microscope (FEI Company, USA): Updated to FEI Tecnai G2 Electron Microscope (FEI Company, Hillsboro, OR, USA).
- Department of Primary Industries and Regional Development (DPIRD) Diagnostics and Laboratory Services (DLS) (South Perth, WA): Updated to Department of Primary Industries and Regional Development (DPIRD) Diagnostics and Laboratory Services (DLS) (South Perth, WA, Australia)
- MasterPure™ DNA Purification Kit (Epicentre Biotechnologies, Madison, Wisconsin, USA): Updated to MasterPure™ DNA Purification Kit (Epicentre Biotechnologies, Madison, WI, USA).
- Kapa Taq (Kapa Biosystems, USA): Updated to Kapa Taq (Kapa Biosystems, Wilmington, MA, USA).
- Australian Genome Research Facility (Perth, WA): Updated to Australian Genome Research Facility (Perth, WA, Australia).
Comments 16: L160-L162: What are “BUN” and “LDH”.
Response 16: Thank you for pointing this out. The abbreviations BUN and LDH have been clarified as follows:
L172 and L176"... lactate dehydrogenase (LDH)…blood urea nitrogen (BUN)..."
Comments 17: L220: Bidirectional sequencing is recommended to get long sequences suitable for phylogenetic analysis.
Response 17: This question has been addressed in Response 28.
Comments 18: L273: “Student’s t-test”, where “t” prior to “-test” in italic.
Response 18: Thank you for highlighting this. The text has been updated as follows:
L304-L305 “For numerical data, the Anderson-Darling test was used to assess normality, followed by Student’s t-test (parametric data).”
Comments 19: L276: “… adjustment, respectively.” “A p value of less than 0.05 is considered statistically significant.” “p” before “value” in italic. Inclusion of 0.05 or not is not clear by the current expression.
Response 19: Thank you for pointing this out. The text has been updated for clarity and to address your comment as follows:
L310-L311 “A p-value of <0.05 was considered statistically significant.”
Comments 20: L295-L307: Sizes of gamonts of different genotypes are different? Microphotographs of gamonts give such impression. Could you provide four photographs for Figures w and 3? Then double the size of figures by arrangement of two by two to show clearly the parasites. This comment is partially related to the comment #34.
Response 20: New photographs focusing in on the parasites have now been provided for Figure 2 and 3 with the addition of small and large macrogamont stages for each genotype. The following has also been added to the Discussion:
L574-579 “Morphologically the Haemocystidium sp. genotypes were polymorphic in nature with different sized immature, macrogamonts and microgamonts detected. Such variation in life cycle forms makes it difficult to distinguish between the different genotypes based on morphology alone. However, general features between the genotypes were noted.”
Comments 21: L304: “single”, not “singular”
Response 21: Thank you for the correction. The text has been updated to read:
L343-L344 “Two gamonts within a single erythrocyte were observed”.
Comments 22: L312: “microgamont” (?)
Response 22: Thank you for picking up the typological error. Although we have changed the Figures, we have ensured the text reads “microgamont” when referring to the pink oblong structure. The Figure 2 (B) caption now reads:
L351-L353 “(B) Oblong light pink microgamont of Haemocystidium sp. with granules scattered and clumped together within the parasite from C. mydas.”
Comments 23: L331-L335 [Figure 4]: It is better to focus on the parasite (Lower half of the erythrocyte shown in the current photograph). Contents of a gamont and a vacuole in the erythrocyte seem to be similar. How the authors differentiate them to be two organisms, not fragments of a single orgamism?
Response 23: We agree with Reviewer 1 that it is difficult to distinguish the parasite stage present and have updated the manuscript and Figure 4 caption as follows:
L371-L375 “Transmission Electron Microscopy (TEM) was undertaken to further investigate the ultrastructural details of the novel parasite (Figure 4). TEM confirmed the presence of intracellular parasitic stages containing electron-dense material and intact membranes within the erythrocyte.”
L379-L382 “Figure 4. Transmission Electron Microscopy (TEM) of novel Haemocystidium sp. parasite within an erythrocyte of a Natator depressus showing electron-dense intracellular parasitic stages (arrows).”
Comments 24. L336-L348: How about infection intensity? No information. It must be closely related to health condition, or health condition assessment. Differences between haemochemical parameters of heavily infected and lightly to none infected individuals could be analysed.
Response 24: This has been addressed in Response 31.
Comments 25: L353-L361: “p” (represent as p-value) in italic.
Response 25: Thank you for pointing this out. The text has been updated to read:
L399-L400 “...foraging turtles were more likely to be positive than nesting turtles (p = 0.004).”
Comments 26: L364-L366: In the section 2.4, numerous parameters for health assessment are shown. But, in the result section, nothing
Response 26: Thank you for your comment. Unfortunately, we had a disparate dataset comprising 20 juvenile green turtles, and 76 foraging flatbacks (including 6 immature), 29 nesting flatback turtles and 5 olive ridley turtles (including 3 immature) that required separate species analysis to assess associations of blood parameter analyses with Haemocystidium presence. A small sample size in green turtles prevented blood parameter analysis (i.e., negative n=3 and positive n=17). Further, with only 2 turtles with high grade Haemocystidium infection, grade-based analyses could not be performed. Thus health assessment parameters were primarily used to classify turtles as healthy or unhealthy. We have added additional exclusion criteria to the Methods, and explicitly addressed the impact of small sample size on our analyses in the Results. We also discuss the abnormal clinical findings in the turtles classed as unhealthy with and without Haemocystidium sp. The single unhealthy turtle positive for Haemocystidium sp., has also been marked in Table 3 with further discussion about the findings in the Discussion. Specifically:
L306-L307 “These numerical analyses were restricted to groups with sample size ≥5.”
L411-L424 “With regards to health, no statistically significant associations were found between the novel Haemocystidium sp. presence or grade and health status or body condition in the foraging group. Of the 130 turtles examined, 13 were classed as ‘unhealthy’, including one turtle positive for Haemocystidium sp. via molecular testing. The turtle exhibited extensive keratin loss and bone exposure of the carapace with epithelialisation at the periphery of the lesion, mild anaemia (PCV 18 L/L, Hb 46 g/L), and elevated AST (486 U/L). The remaining unhealthy turtles, all negative for Haemocystidium sp., presented with external abnormalities such as lacerations, exudative carapace and plastron lesions, dehydration, heavy barnacle burdens and a range of clinically significant blood alterations, including severe anaemia (PCV 6 L/L), hypoalbuminaemia (6 g/L), leukocytosis (49.9 x 109 cells/L), heterophilia (30.94 x 109 cells/L), eosinophilia (8.88 x 109 cells/L), and elevated GLDH (318 U/L).”
L431-L440 “In the foraging N. depressus blood analysis, no significant differences were observed in any haematological or biochemical parameters between the positive and negative cases, including in the mature foraging N. depressus group…. No haemolysis was reported and small sample sizes precluded certain blood parameter analyses including species-specific analyses for C. mydas and grade-based analyses for Haemocystdium sp.”
L536-L542 “Blood parasites can also contribute to anaemia, but the single Haemocystidium-positive unhealthy C. mydas with anaemia also had chronic carapace lesions and elevated AST, suggesting tissue damage (Stacy and Innis, 2017). Hence, the observed anaemia could not be definitively attributed to parasite presence.”
Comments 27: L379, L380: Refer to Comments 6 and 11.
Response 27: Thank you for the observation. As mentioned under Response 11, the term cytochrome b gene (cytb) has been updated here and throughout the manuscript for consistency.
Comments 28: L380-LL383: Mention about intra-group (genotypes in the present version of manuscript) nucleotide sequence variation. Absolutely identical? Recommend again bi-directional sequencing of all or selected (at least three isolates per genotype).
Response 28: Thank you for pointing this out. Bidirectional sequencing was performed for a representative of genotype A T154 and exhibited a 100% homology to T195 which was selected for phylogenetic analysis in this study. All genotype A isolates from the sea turtles were aligned together and exhibited 100% similarity to each other. Bidirectional sequencing was performed for two representatives of genotype B, T212 and T204 which exhibited a 100% homology to T239 which was selected for phylogenetic analysis in this study. The text has been updated and now reads:
L239-L241 “unidirectional sequencing with a selection of isolates chosen for bi-directional sequencing.”
Comments 29: L387: “nominal species” (?)
Response 29: The term ‘name’ has now been replaced with ‘nominal’ and reads as follows:
L453 “nominal species”.
Comments 30: L425-L427: Closing parentheses are lacked? Or, remove unnecessary starting parenthesis.
Response 30: Thank you for pointing this out. The unnecessary starting parenthesis has been removed. The revised sentence now reads:
L497-L499 “When comparing sea turtle Haemocystidium sp. genotype A to known described Haemosporidia species, Haemoproteus anatolicum isolate 5315 (KM068153) and Haemoproteus caucasica isolate 4111 (KM015455).”
Comments 31: L449-L451: In this manuscript, infection intensity is not shown. Similarly, any changes of haemochemical parameters are mentioned. It is unclear how the authors concluded their results as no association between Haemocystidium infection and health. Intra-erythrocytic stage and extra-erythrocytic stage of the haemoparasite may affect host health. How do the authors understand Hamocystidium infection and its impact to the host? It is unclear in this manuscript.
Response 31: Thank you for your comments and for highlighting these areas of concern. To address the Reviewer’s question regarding infection intensity, Haemocystidium grade for the positive microscopy cases has been added to Table 3. For additional clarity, WTR211, the only Haemocystidium-positive turtle classified as ‘unhealthy’ has been identified, and the updated caption now reads:
L425“Table 3. Sea turtle novel Haemocystidium sp. positive microscopy and/or PCR, including demographics. All were foraging turtles (* WTR211 was the only turtle classed as ‘unhealthy’; the rest were classed as healthy).”
Additionally, the Methods section has been updated to describe how infection intensity analysis was performed, stating:
L279-L283 “The Haemocystidium grade analysis was based solely on light microscopy results, where severe (3+) and moderate (2+) were combined (assigned ‘high’ grade), and mild (1+) and absent (0) were combined (assigned ‘low/absent’ grade)..”
Regarding the presence of extra-erythrocytic stages, none were detected during our study. To highlight this, the following sentence has been added to the results section:
L317-L318 “No extraerythrocytic stages of the haemoparasite were observed”.
To address the Reviewer’s concern regarding the conclusion about Haemocystidium sp. and its impact on health, we note that no statistically significant correlations were observed with blood analytes in N. depressus. We have updated the corresponding paragraph to emphasise this lack of significant associations, to ensure clarity and avoid misinterpretation. Please also see Response 26 for further information.
L431-L440 “In the foraging N. depressus blood analysis, no differences were observed in any of the blood values between the positive and negative cases, including in the mature foraging N. depressus group…. All other haematological and biochemical parameters showed no significant associations with Haemocystidium sp. presence and no haemolysis was reported. Small sample sizes precluded certain blood parameter analyses including species-specific analyses for C. mydas and grade-based analyses for Haemocystdium sp.”
To further address the Reviewer’s concern regarding how we assessed health, we have added further details to the Methods section regarding how blood values were assessed:
L267-L269 “Health status was determined through a combination of physical examination findings and blood results by comparing to species-specific blood reference intervals (Flint et al., 2010; Whiting et al., 2007a; Young et al., 2024).”
Additionally, we acknowledge that the limited sample size may influence our ability to detect associations. To address this, we have included additional information in the Limitations and Recommendations section about the need for larger sample sizes to refine our understanding of Haemocystidium sp., including its effects on hosts, the role of infection phase, and its relationship with health parameters. The updated section reads:
L624-L635 " While larger samples sizes, including at nesting beaches, are needed to better assess parasite presence and other factors, smaller datasets, often unavoidable in endangered species studies, can provide valuable exploratory insights (Bissonette, 1999). ... Larger studies, including naïve hosts, are essential to clarify the effects of Haemocystidium sp. on hosts, the role of infection phase in parasite dynamics, and the lack of correlation with health parameters.”
Comments 32: L463-L465: No information about haemolysis evaluation is given in the results section, but here the authors give it.
Response 32: Thank you for your observation. To address this, the following statement has been added to the results section:
L438“No haemolysis was observed.”
Comments 33: L481-L488: Acute infection phase? Chronic infection phase?
Response 33: Thank you for this comment. Additional information about infection phase and need for larger sample sizes has been added to the text and now reads:
L562-L567“Contributing factors may include species-specific differences in immune responses, vector exposure, ecological behaviour, or infection phase. For example, low levels of circulating parasites in N. depressus may reflect a chronic phase driven by coevolution and host adaptation, while higher levels in C. mydas could indicate an acute phase (Nardini, 2019; Telford, 2009; Zajac et al., 2021).
L630-635 “Larger studies, including naïve hosts, are essential to clarify the effects of Haemocystidium sp. on hosts, the role of infection phase in parasite dynamics, and the lack of correlation with health parameters.”
Comments 34: L489-L508: Discuss morphological differences of species A and B, which could support molecular differences between them, after writing information in the Results section.
Response 34: As previously mentioned, the Haemocystidium genotypes A and B are not considered separate species as a genetic distance of of 5% or greater at the cytb locus is required for species delimitation. Both genotypes showed similar morphological stages with immature, microgamonts and macrogamonts of various sizes identified in erythrocytes. Morphological measurements and size range of each of the different life-cycle stage would be required to identify any differences and or similarities between the sea turtle Haemocystidium sp. when larger data sets become available.
Comments 35: L509-L528: It is unclear whether the authors want to discuss about speciation of the species (evolution) or ecological relationships between the species and their distribution.
Response 35: Thank you for pointing out the need for clarity in this section. The focus of the discussion of this paragraph is primarily on speciation and evolutionary relationships, as demonstrated by the phylogenetic analysis of Haemocystidium genotypes A and B, which supports their placement within the reptilian clade and their monophyletic grouping. To clarify, we reviewed the ecological reference to more closely align with the evolutionary discussion to read as follows:
L608-613 “The restricted geographic range of N. depressus, along with ecological and host-specific factors, may have influenced the genetic divergence and prevalence of genotype B, highlighting the interaction between evolutionary and ecological processes.”
Comments 36: L542-L544: The parasite studies here is a protozoon. It is necessary why the immature stage has limited DNA compared to the developed stage by a reference.
Response 36: The authors agree with Reviewer 1 and have removed the following statement:
L638-L640 “It is possible that these false negatives may be attributed to limited DNA in the immature ring stage in these cases.”
The new statement now reads:
L640-L644 “It is possible that DNA amplification failed due to low grade parasitemia, reducing the concentration of the starting DNA template, or, conversely, high concentrations of DNA inhibited PCR amplification, potentially due to the presence of nucleated red blood cells in the sample”.
Comments 37: L18-L578: To clean-up the study results and discussion of the present study, it is better to re-organize the contents, discard unnecessary descriptions and discussion, and add description of important data. This is significant study, and its improvement increases the scientific value of this study.
Response 37: Thank you for your thoughtful feedback. We’ve made significant revisions to improve the manuscript, focusing on reorganizing the Results and Discussion sections to make them clearer and more concise. Unnecessary details have been removed, and we’ve added important data, such as the results of blood parameter analyses, to strengthen the study’s findings. We’ve also addressed the limitations caused by small sample sizes, which restricted some analyses, including species-specific and grade-based associations with Haemocystidium sp. Additionally, we’ve clarified why we classified the two Haemocystidium genotypes as genotypes rather than separate species, using established genetic distance thresholds and evidence from previous studies. The discussion has been expanded to include abnormal clinical findings in unhealthy turtles and to highlight the lack of statistically significant associations between Haemocystidium sp. and host health. We believe these changes address your concerns and have greatly improved the clarity and impact of the study (L18-L678).
Comments 38: L680: Remove “-104576”.
Response 38: Thank you for pointing this out. The citation has been updated to remove duplicate number:
L796-L798 “Pacheco, M. A., L. M. P. Ceríaco, N. E. Matta, M. Vargas-Ramírez, A. M. Bauer, and A. A. Escalante. 2020. "A phylogenetic study of Haemocystidium parasites and other Haemosporida using complete mitochondrial genome sequences." Infection, Genetics and Evolution 85: 104576”
Comments 39: L690: “343-344”
Response 39: Thank you for picking this up. The citation has been updated to correct the page number:
L812-L813 “Stacy, N. I., and J. W. Harvey. 2015. "Letter to the Editor: Heinz bodies and erythrophagocytosis in the peripheral blood of loggerhead sea turtles." Journal of Experimental Zoology. Part A, Ecological Genetics and Physiology 323 (6): 343-344.”

Reviewer 2 Report
Comments and Suggestions for Authors
This study provides valuable and novel information on the discovery of novel hemoparasite in sea turtles flatback (Natator depressus) and green (Chelonia mydas) turtle erythrocytes during routine blood film examination from Australian coasts. By molecular assessment, a novel Haemocystidium sp. was identified. The authors tested the effect of presence of such parasite in the examined turtle using morphometeric, clinical, and laboratory analyses and no significant effect was observed. The molecular and phylogenetic analysis was conducted for confirmation of results. The manuscript is well-written and no serious issues were detected for language or writing style. However, some additional information or inquiries need clarification from the authors as illustrated in below.
- I cannot find any data or figures related to clinical or laboratory investigations in the submitted file of reviewing, although they were reported in the methodology (Lines 106-172). Detailed information of such assessments is highly required in such type of preliminary study.
- I recommend addition of some information on the clinical findings or pathogenesis of such identified protozoa even from any closely related protozoa or hosts in the introduction section.
Author Response
This study provides valuable and novel information on the discovery of novel hemoparasite in sea turtles flatback (Natator depressus) and green (Chelonia mydas) turtle erythrocytes during routine blood film examination from Australian coasts. By molecular assessment, a novel Haemocystidium sp. was identified. The authors tested the effect of presence of such parasite in the examined turtle using morphometeric, clinical, and laboratory analyses and no significant effect was observed. The molecular and phylogenetic analysis was conducted for confirmation of results. The manuscript is well-written and no serious issues were detected for language or writing style. However, some additional information or inquiries need clarification from the authors as illustrated in below.
Comments 1: I cannot find any data or figures related to clinical or laboratory investigations in the submitted file of reviewing, although they were reported in the methodology (Lines 106-172). Detailed information of such assessments is highly required in such type of preliminary study.
Response 1: Thank you for your comment. Unfortunately, due to a disparate dataset (20 juvenile green turtles, and 76 foraging flatbacks (including 6 immature), 29 nesting flatback turtles and 5 olive ridley turtles (including 3 immature) and that required separate species analysis to assess associations of blood parameter analyses with Haemocystidium presence created small sample size issues. A small sample size in green turtles prevented blood parameter analysis (i.e., negative n=3 and positive n=17). Further, with only 2 turtles with high grade Haemocystidium infection, grade-based analyses could not be performed. Thus health assessment parameters were primarily used to classify turtles as healthy or unhealthy. We have added additional exclusion criteria to the Methods, and explicitly addressed the impact of small sample size on our analyses in the Results. Further, to address the Reviewer’s comments, we also discuss the abnormal clinical findings in the turtles classed as unhealthy with and without Haemocystidium sp. The single unhealthy turtle positive for Haemocystidium sp., has also been marked in Table 3 with further discussion about the findings in the Discussion.
L306-L307 “These numerical analyses were restricted to groups with sample size ≥5.”
L411-L424 “With regards to health, no statistically significant associations were found between the novel Haemocystidium sp. presence or grade and health status or body condition in the foraging group. Of the 130 turtles examined, 13 were classed as ‘unhealthy’, including one turtle positive for Haemocystidium sp. via molecular testing. The turtle exhibited extensive keratin loss and bone exposure of the carapace with epithelialisation at the periphery of the lesion, mild anaemia (PCV 18 L/L, Hb 46 g/L), and elevated AST (486 U/L). The remaining unhealthy turtles, all negative for Haemocystidium sp., presented with external abnormalities such as lacerations, exudative carapace and plastron lesions, dehydration, heavy barnacle burdens and a range of clinically significant blood alterations, including severe anaemia (PCV 6 L/L), hypoalbuminaemia (6 g/L), leukocytosis (49.9 x 109 cells/L), heterophilia (30.94 x 109 cells/L), eosinophilia (8.88 x 109 cells/L), and elevated GLDH (318 U/L).”
L431-L440 “In the foraging N. depressus blood analysis, no differences were observed in any of the blood values between the positive and negative cases, including in the mature foraging N. depressus group…. All other haematological and biochemical parameters showed no significant associations with Haemocystidium sp. presence and no haemolysis was reported. Small sample sizes precluded certain blood parameter analyses including species-specific analyses for C. mydas and grade-based analyses for Haemocystdium sp.”
L537-L542 “Blood parasites can also contribute to anaemia, but the single Haemocystidium-positive unhealthy C. mydas with anaemia also had chronic carapace lesions and elevated AST, suggesting tissue damage (Stacy and Innis, 2017). Hence, the observed anaemia could not be definitively attributed to parasite presence.”
Comments 2: I recommend addition of some information on the clinical findings or pathogenesis of such identified protozoa even from any closely related protozoa or hosts in the introduction section.
Response 2: Thank you for your suggestion. Additional information on the clinical findings and pathogenesis of haemosporidians has been added to the introduction for context. The updated text includes:
L65-L73 “Haemosporidia (Apicomplexa) are a diverse group of parasites which infect a wide range of vertebrate hosts including birds, reptiles and mammals (Lainson and Naiff, 1998; Maia et al., 2016). This group includes Plasmodium spp., the causative agent of malaria and a parasite of critical One Health significance and ecological importance (Valkiūnas and Iezhova, 2018; Wellehan and Walden, 2019). These parasites are typically transmitted by dipteran vectors and have indirect and complex life cycles, predominantly intraerythrocytic but also involving extraerythrocytic stages (Nardini, 2019; O'Donoghue, 2017).”
L89-L92 “For instance, blood parasite infections in reptiles, may lead to anaemia, haemolysis and splenomegaly, particularly under conditions of stress or immunosuppression (Nardini, 2019).”
Round 2
Reviewer 2 Report
Comments and Suggestions for Authors
The authors revisions have been improved the quality of manuscript markedly and manuscript can be accepted.